# Perceptions of Chinese Towards Dementia in Hong Kong—Diagnosis, Symptoms and Impacts

**DOI:** 10.3390/ijerph16010128

**Published:** 2019-01-05

**Authors:** Tai Pong Lam, Kai Sing Sun, Hoi Yan Chan, Chak Sing Lau, Kwok Fai Lam, Robert Sanson-Fisher

**Affiliations:** 1Department of Family Medicine and Primary Care, The University of Hong Kong, Hong Kong, China; kssun2@hku.hk (K.S.S.); step0826@hku.hk (H.Y.C.); 2Department of Medicine, The University of Hong Kong, Hong Kong, China; cslau@hku.hk; 3Department of Statistics and Actuarial Science, The University of Hong Kong, Hong Kong, China; hrntlkf@hku.hk; 4School of Medicine and Public Health, The University of Newcastle, Callaghan 2308, Australia; rob.sanson-fisher@newcastle.edu.au

**Keywords:** Chinese, dementia, diagnosis, knowledge, perception, symptoms

## Abstract

The increasing prevalence of dementia has become a public health issue worldwide including China. This study aims to explore the perception of Chinese in Hong Kong towards the diagnosis, symptoms and impacts of dementia. A cross-sectional survey was conducted among outpatients (without diagnosed dementia) attending a regional public hospital using a standard questionnaire. The results from 290 respondents showed that most preferred to be told about the diagnosis of dementia as soon as possible if they got it, in order to deal with the news and to access treatment and support early. Nearly two thirds of the respondents perceived practical issues (61.3%), physical health (61.0%), and emotional distress (58.4%) as their most fearful impacts, while legal issues (7.4%) were their least concerns. Family history/genes (79.1%) and brain injury (75.9%) were the most commonly perceived causes of dementia. For symptoms, respondents were more likely to identify cognitive impairments than undesirable behaviours. The accepting and proactive attitudes of the public indicate that there is a timely need of more public education about the disease, early screening and better continuity of care to fulfil the anticipated increase of the dementia patient population.

## 1. Introduction

With the ageing of the population worldwide, the increasing prevalence of dementia has become a global public health issue [1]. Dementia is now the leading cause of death in the UK [2]. It also ranks the first, second and sixth cause of disability among the elderly in the UK [2], Australia [3] and the US, respectively [4]. In Hong Kong, approximately 9.3% of adults aged 70 or above are affected by dementia [5,6]. A significant increase in local prevalence is highly likely as Hong Kong population enjoys the world’s highest life expectancy [7,8]. According to the Census and Statistics Department, the proportion of elderly aged 65 or over in Hong Kong is going to double in 25 years, from 16.5% in 2017 to 31.0% in 2042 [9]; citizen aged 85 or older was 2.5% in 2017 and is projected to increase more than double to 6.7% by 2042.

Dementia, a chronic syndrome characterized by progressive decline of cognitive function that interferes independence in activities of daily living, often has a profound impact on not only the person diagnosed and their immediate family [10], but also their extended family and social network [11,12]. Currently, no cure but pharmacological and non-pharmacological interventions that help delay the functional deterioration for dementia are available. Evidence to support the use of pharmaceutical agents or dietary supplements in preventing Alzheimer’s had remained insufficient [13]. Some pharmacological interventions, for example acetylcholinesterase inhibitors, are found to have better efficacy in ameliorating symptoms in the early stages of dementia. There is limited evidence for non-pharmacological interventions for people with dementia to effectively slow decline and delay institutionalisation. However, psychoeducation for caregivers that started earlier in the disease course could be effective in improving caregivers’ mood and quality of life. For maximum intervention options and benefits, timely detection and identification of dementia becomes the key. Evidence revealed that early recognition of dementia could reduce psychological distress of both patients and their families [14,15,16], while advance planning and persistent treatment during the mild-to-moderate stages of dementia could slow down the progression of cognitive decline [17,18,19,20].

Community perceptions of dementia is mixed. One meta-analysis reported that some patients and their families were very distressed about the possibility of dementia. Their fear and reluctance to face the disease were attributed to delaying the timing of diagnosis [21]. In other studies, patients were generally agreeable to dementia screening, especially if they understood the benefits of early identification [22,23,24,25]. A recent study in Australia suggested that people rather wished to be told about the diagnosis as soon as possible, regardless of demographics or any personal relation to dementia patients [26]. Meanwhile, data on the reception of Chinese towards a diagnosis of dementia relative to diagnoses of other health conditions, and the reasons for their reactions, are scarce apart from a Singaporean study in which people generally showed reluctance and avoidance towards the disease and screening [27]. Cultural difference regarding moral and familial values between Westerners and South East Asians could be a crucial factor that predicts variations in attitudes and the level of readiness towards screening initiatives and timely diagnosis [28].

As a consequence of the staggering ageing rate, dementia will inevitably put a heavy burden on the social and healthcare system in Hong Kong. There is growing awareness for the need to raise public literacy and to plan effective service deliveries in this respect. This study aims to explore the perception of Chinese in Hong Kong towards the diagnosis, symptoms and impacts of dementia. The findings will inform policy makers and healthcare providers about the need for public education to reduce stigma associated with the condition, and may serve as reference on how best to allocate resources to reduce the impact of dementia.

## 2. Materials and Methods

### 2.1. Questionnaire

We adopted a questionnaire developed by the Australian research collaborators [26,29]. The questionnaire asked respondents about their fear of dementia compared with other diseases, causes and symptoms of dementia, timing of diagnosis disclosure and impact of dementia. The original questionnaire was written in English. It was forward translated into Chinese and then back translated into English. The backward translation was cross-checked with the original English version to ensure preserved meaning and consistent content across the English and Chinese versions. The final questionnaire was pilot-tested for its face- and content-validity with laymen. Most of the items were rated as comprehensible and relevant.

### 2.2. Sample

A cross-sectional survey was conducted among hospital outpatients from an internal medicine department of a regional public hospital during May 2018. The target population was Chinese patients aged 18 or over, able to read and speak Chinese, capable of providing informed consent and completing a survey. Patients who were not physically or mentally capable to participate or had an existing diagnosis of dementia were excluded. Patients at the clinic waiting area were invited by research assistants to complete the questionnaire. Most participants completed the questionnaire by themselves. For some elderly participants who had difficulties in reading, the research assistants helped to administer the questionnaire. To encourage responses, HK$20 (US$2.6) was offered to each respondent as incentive. Ethics approvals were obtained from the Institutional Review Board of The University of Hong Kong/Hospital Authority Hong Kong West Cluster (UW 18-194). Written consent was obtained from the participants.

### 2.3. Statistical Analysis

The survey data were analysed using IBM SPSS Statistics for Windows, version 24 (IBM Corp., Armonk, NY, USA). Pearson Chi-squared test was used to test for the differences in the responses among independent groups, while McNemar test was used for paired groups. A *p*-value < 0.05 was considered statistically significant.

## 3. Results

### 3.1. Participants Recruited

Excluding 6 incomplete questionnaires (major sections unanswered) and 2 responses from patients diagnosed with dementia, 290 respondents successfully completed the questionnaires, with a response rate about 75% for the eligible subjects. Of the respondents, 46.5% were males and 53.5% were females. Their age ranged from 18–91, with a mean 53.2 (13.24) years. They were grouped into three age-groups (<40, 40–64, ≥65 years old) in the following analyses.

### 3.2. Fear of Dementia

Table 1 compares the rating of the most feared disease among different age groups. Fear of dementia increased with age, dementia was rated as the most feared disease (27.5%) by the respondents aged ≥65 compared to 14.8% of those aged 40–64 and 7.3% of the aged <40 group. While cancer was the second most feared disease among the oldest group (22.5%), it was rated the first among the younger groups (49.1% and 70.7%, respectively).

### 3.3. Perceived Impact of Dementia

Respondents were asked to select up to three impacts which they found the most difficult to deal with if they or their significant others had dementia. (Table 2). More than half rated practical issues (e.g., needing assistance with daily tasks, unable to drive) (61.3%) and physical health (61.0%) as the problems with the most impact if they themselves had dementia, followed by emotional impact (58.4%), financial strain (38.7%) and social impact (32.7%). Significant differences were shown by McNemar test between the impacts on oneself and significant others regarding the social (<0.001), practical (0.003) and legal (0.049) aspects. If the case happened to their significant others, respondents would have less concern about the social impact (18.2%) and legal issue (4.1%) associated with dementia but would perceive stronger impact from practical issues (71.4%).

We further compared the attitudes towards impact of dementia on significant others between the respondents knowing someone diagnosed with dementia in the past 5 years and those who did not know (Table 3). The former had less concern about financial impact (25.4% vs. 45.5%; *p* = 0.004) but perceived greater problems about dealing with the health system (37.3% vs. 21.3%, *p* = 0.009).

### 3.4. Preferences for the Timing of Diagnosis Disclosure

Given there is no cure, respondents were asked about the timing of knowing the diagnosis of dementia. The majority (87.4%) wanted to know it as soon as possible. This group of respondents were further asked about their reasons for early diagnosis. The top three reasons were: access treatments and support (64.9%), come to terms with the diagnosis (57.3%), be involved in decisions about my future care (53.8%). Other reasons included: make the most of life (e.g., bucket list) (47.4%), tell loved ones my situation (42.7%), make financial arrangements with family (39.8%), work on my relationships (28.1%), and collect memories (24.0%).

### 3.5. Causes of Dementia

Combining “agree” and “strongly agree” responses, most respondents regarded family history or genes (79.1%) and brain injury (75.9%) as the causes of dementia, followed by stress or worry, (71.5%) and medications (67.1%). Over half of the respondents also agreed with lack of exercise (61.1%), and drinking too much alcohol (51.6%) as the causes. It is noted that up to one third of the respondents were unsure about the answers, especially for the factors about chemicals in the room (36.2%), food additives/preservatives (33.8%), high blood pressure (32.6%), pesticides (32.3%), smoking (30.6%), poor diet (30.4%) and overweight (30.3%). Details of the responses are shown in Table 4.

Pearson Chi-squared test was conducted to analyse the association between the responses and age. As the number of “strongly disagree” responses were small, they were combined with the “disagree” responses in the analysis. We found that older respondents were more likely to agree with poor diet as the cause of dementia.

### 3.6. Symptoms of Dementia

Over 90% of the respondents agreed/strongly agreed that trouble recognising family members (93.5%), wandering away from home (92.8%) and trouble remembering recent events (92.7%) were the symptoms of dementia. Besides, most regarded losing or misplacing things (85.4%) and forgetting what day it is (81.0%) as the symptoms, followed by needing assistance with everyday tasks (72.7%) and trouble finding the right word or misnaming things (67.2%). It is noted that over one quarter of the respondents were unsure about whether showing inappropriate sexual behaviours (31.1%), clingy behaviour (31.1%), being aggressive towards others (26.5%) and talking loudly and rapidly (26.4%) were the symptoms of dementia. Details are shown in Table 5.

Again, Pearson Chi-squared test was conducted to analyse the association between the responses and age. “Strongly disagree” responses were combined with the “disagree” responses in the analysis. We found that younger respondents were more likely to agree with the followings as the dementia symptoms: trouble remembering recent events, trouble recognising family members, losing or misplacing things, forgetting what day it is and wandering away from home.

## 4. Discussion

While the benefits of early detection and diagnosis of dementia have been recognized by doctors and researchers for better patient and caregiver outcomes, it is important to understand the general public’s perception and reception of the disease which will be helpful in determining how to facilitate education and service provision in the community. The current study examined the perception and attitudes towards dementia and associated issues among the Chinese in Hong Kong. Similar to the data in a 5-country survey [30], dementia was rated not as fearful as cancer among younger adults. However, the fear of dementia was found to become worse than that of cancer or stroke with increasing age. With age being a well-recognized predictor of dementia, such anxiety could also be due to cohort effect that they are more likely than younger adults to meet peers with negative ageing stereotypes [31].

From our findings, the most worrying perceived impacts for oneself and significant others substantially reflected the perceived stress and anxiety as a patient or a carer. Regardless of knowing someone with dementia or not, physical sufferings, the inability to perform activities of daily living and emotional impacts rated top 3 for both of the roles. For the carers, however, the loss of ‘independence’ in daily activities, rather than the loss of friends or hobbies for the patients was of greater concern to them. It was also noteworthy that most of our respondents did not recognize the display of aggressive behaviours and inappropriate sexual behaviours as possible symptoms, reflecting not only a knowledge gap in this respect among the local population, but also a delay in seeking timely help for behavioral disturbances or psychotic symptoms the deterioration of which would render community care untenable [32]. Apart from the loss of identity and emotional vulnerability of the patient, agitated behaviours and tantrums from the patients could further distress the caregiver [33,34,35]. Such behavioural and psychological symptoms of dementia (BPSD) seemed to burden Asians, especially Chinese, carers more than others [36,37]. Research findings on carers for patients with mental illness suggested that Hong Kong Chinese had even more caregiving burdens than other Chinese counterparts despite having better medical facilities and services, which could probably be due to the city’s specific set of values of mixed Capitalism and Confucianism [32]. Remarkably, there was evidence that BPSD could be the most manageable aspect of the disease, with a range of treatment options, if identified early [38,39].

Almost 90% of the respondents in our study preferred to be told as soon as possible if they were diagnosed with dementia, and 80% when it came to considering for the partner or spouse. This is consistent with the Australian finding of 92% [26]. The majority of our respondents expressed the wish to have more time to come to terms with the diagnosis as well as to get access to adequate treatment and reliable support. In other words, acknowledging the benefits of early diagnosis might ease some concerns and distress prompted by the diagnosis, and also facilitate primary care doctors to communicate and manage patients who suffer from dementia.

Another notable finding from our study was that people who knew someone with dementia were more likely to express concern over dealing with the healthcare system. This might have possibly revealed how ill-equipped our city is towards ageing and the increasing prevalence of dementia. In Hong Kong, the legal and medical gaps together with the lack of a well-thought out plan for dementia care might easily lead to confusion and helplessness. Little community-based intervention, with evidence improving both patient and caregiver outcomes [40], is available. In addition, the tendency that Hong Kong people only seek medical attention at late stages has not been a facilitator in fulfilling the demand for dementia care [32]. Perhaps due to easy dismissal of the symptoms at the early stages of dementia, patient and family often remained unwary until advanced decline in spite of their accepting and proactive attitude towards an early diagnosis of dementia. Apart from the need to improve education and awareness of dementia and advanced care planning in the community, it is also important for primary care practitioners to enhance their competence and thus readiness to engage in a more active role in managing dementia [41]. Broader implications include advising the government and policy makers to address the inadequacies in current dementia care by enhancing the socio-medical support for both patients and their families throughout the dementia trajectory.

There were limitations in this study. First, the survey respondents were hospital outpatients. The results might not be generalizable to the general public in Hong Kong. Second, some questions were hypothetical, the patients might have different perceptions of the impact of dementia if they encountered it. Patients who had an existing diagnosis of dementia or not being physically or mentally capable to answer the questionnaire were excluded in the survey.

## 5. Conclusions

The results of current study demonstrated that Hong Kong Chinese preferred to be told about their diagnosis. They also wished to receive the disclosure as soon as possible in order to deal with the news and to access treatment and support early. Nearly two thirds of the respondents perceived practical issues, physical health, and emotional distress as the most fearful impacts, while legal issues were their least concerns. Family history, genes and brain injury were the most commonly perceived causes of dementia. For symptoms, respondents were more likely to identify cognitive impairments than undesirable behaviours. The accepting and proactive attitudes of the public could be an indication to policy makers that there is a timely need of more public education about the disease and a structured pathway of dementia care for early screening and better continuity of care to fulfil the anticipated increase of the dementia patient population.

## Figures and Tables

**Table 1 ijerph-16-00128-t001:** Comparison of the most feared disease among different age-groups (select 1 item).

Chronic Disease	<40 *n* = 41	40–64 *n* = 169	≥65 *n* = 40
	%	%	%
Coronary heart disease (not high blood pressure or high cholesterol)	2.4%	0.6%	2.5%
Dementia and Alzheimer’s disease	7.3%	14.8%	27.5%
Cerebrovascular disease (such as stroke)	14.6%	23.7%	17.5%
Cancer	70.7%	49.1%	22.5%
Chronic obstructive pulmonary disease (such as emphysema)	0.0%	0.6%	0.0%
Diabetes	0.0%	1.8%	5.0%
Heart failure	4.9%	1.8%	10.0%
None of the above	0.0%	4.7%	12.5%
Others	0.0%	3.0%	2.5%

**Table 2 ijerph-16-00128-t002:** Perceived impact of dementia to self and significant others (select up to 3 items).

Perceived Impact of Dementia	Self		Significant Others		McNemar Test
	*n*	%	*n*	%	*p*-Value
Emotional impact (e.g., stress, loss of identity)	157	58.4%	157	58.4%	1.000
Social impact (e.g., loss of friends or hobbies)	88	32.7%	49	18.2%	<0.001 **
Practical issues (e.g., needing assistance with daily tasks, unable to drive)	165	61.3%	192	71.4%	0.003 *
Financial strain (e.g., difficulty paying for medical expenses)	104	38.7%	113	42.0%	0.362
Physical health (e.g., problems with toileting)	164	61.0%	151	56.1%	0.218
Legal issues (e.g., appointing someone to make decisions on my behalf)	20	7.4%	11	4.1%	0.049 *
Dealing with the health system (e.g., getting what I need from health providers)	57	21.2%	69	25.7%	0.162

* *p* < 0.05; ** *p* < 0.001 Percentages refer to valid responses only. The pair of responses would be excluded in the analysis if either the responses regarding self or significant others was missing.

**Table 3 ijerph-16-00128-t003:** Comparison of the perceived impact of dementia to others between respondents knowing someone with dementia in the past 5 years and those did not (select up to 3 items).

Perceived Impact of Dementia	Knew Someone with Dementia		Did Not Know Someone with Dementia		Pearson χ^2^ Test
	*n*	%	*n*	%	*p*-Value
Emotional impact (e.g., stress, loss of identity)	39	58.2%	121	59.9%	0.807
Social impact (e.g., loss of friends or hobbies)	12	17.9%	36	17.8%	0.987
Practical issues (e.g., needing assistance with daily tasks, unable to drive)	49	73.1%	141	69.8%	0.604
Financial strain (e.g., difficulty paying for medical expenses)	17	25.4%	92	45.5%	0.004 *
Physical health (e.g., problems with toileting)	35	52.2%	113	55.9%	0.598
Legal issues (e.g., appointing someone to make decisions on my behalf)	2	3.0%	8	4.0%	0.715
Dealing with the health system (e.g., getting what I need from health providers)	25	37.3%	43	21.3%	0.009 *

* *p* < 0.05.

**Table 4 ijerph-16-00128-t004:** Views on the causes of dementia.

Cause	Strongly Agree		Agree		Disagree		Strongly Disagree		Not Sure		Pearson χ^2^Test with Age Groups
	*n*	%	*n*	%	*n*	%	*n*	%	*n*	%	*p*-Value
(a) Family history or genes	57	20.5%	163	58.6%	16	5.8%	2	0.7%	40	14.4%	0.920
(b) Stress or worry	34	12.5%	161	59.0%	25	9.2%	1	0.4%	52	19.0%	0.083
(c) Smoking	19	7.1%	79	29.5%	76	28.4%	12	4.5%	82	30.6%	0.476
(d) Poor diet	22	8.1%	86	31.5%	71	26.0%	11	4.0%	83	30.4%	−0.048 *
(e) Drinking too much alcohol	21	7.7%	119	43.9%	53	19.6%	5	1.8%	73	26.9%	0.494
(f) Lack of exercise	27	10.0%	138	51.1%	51	18.9%	6	2.2%	48	17.8%	0.424
(g) Being overweight	17	6.4%	64	24.0%	97	36.3%	8	3.0%	81	30.3%	0.357
(h) Medications	32	11.9%	149	55.2%	25	9.3%	3	1.1%	61	22.6%	0.578
(i) Chemicals in the home (e.g., cleaning products)	13	4.9%	73	27.5%	76	28.7%	7	2.6%	96	36.2%	0.181
(j) Food additives/preservatives	14	5.2%	110	40.9%	50	18.6%	4	1.5%	91	33.8%	0.283
(k) Pesticides	18	6.7%	105	39.0%	54	20.1%	5	1.9%	87	32.3%	0.709
(l) Air pollution	17	6.3%	100	37.0%	66	24.4%	9	3.3%	78	28.9%	0.244
(m) Working hours (e.g., long hours, shift work)	26	9.7%	101	37.7%	65	24.3%	5	1.9%	71	26.5%	0.130
(n) High blood pressure	22	8.1%	105	38.5%	53	19.4%	4	1.5%	89	32.6%	0.454
(o) Brain injury (e.g., from a car accident)	53	19.3%	155	56.6%	16	5.8%	1	0.4%	49	17.9%	0.303

* *p* < 0.05. The symbol “–”means the older group was significantly more likely to agree strongly with the item.

**Table 5 ijerph-16-00128-t005:** Views on the symptoms of dementia.

Symptoms of Dementia	Strongly Agree		Agree		Disagree		Strongly Disagree		Not Sure		Pearson χ^2^Test with Age Groups
	*n*	%	*n*	%	*n*	%	*n*	%	*n*	%	*p*-Value
(a) Trouble remembering recent events	95	34.4%	161	58.3%	11	4.0%	0	0.0%	9	3.3%	+, 0.018 *
(b) Trouble recognising family members	104	38.0%	152	55.5%	6	2.2%	5	1.8%	7	2.6%	+, 0.001 *
(c) Losing or misplacing things	73	26.6%	161	58.8%	20	7.3%	3	1.1%	17	6.2%	+, 0.001 *
(d) Forgetting what day it is	66	24.2%	155	56.8%	28	10.3%	4	1.5%	20	7.3%	+, 0.008 *
(e) Destroying property	11	4.1%	66	24.7%	118	44.2%	10	3.7%	62	23.2%	0.054
(f) Talking loudly and rapidly	12	4.5%	65	24.2%	101	37.5%	20	7.4%	71	26.4%	0.092
(g) Being aggressive towards others	12	4.5%	53	19.8%	114	42.5%	18	6.7%	71	26.5%	0.342
(h) Showing inappropriate sexual behaviours	10	3.7%	36	13.5%	115	43.1%	23	8.6%	83	31.1%	0.098
(i) Wandering away from home	102	37.0%	154	55.8%	6	2.2%	3	1.1%	11	4.0%	+, 0.047 *
(j) Needing assistance with everyday tasks (e.g., dressing)	54	19.9%	143	52.8%	42	15.5%	4	1.5%	28	10.3%	0.108
(k) Problems getting to the toilet on time	31	11.4%	114	42.1%	57	21.0%	6	2.2%	63	23.2%	0.640
(l) Lacking interest in personal hygiene or grooming	25	9.4%	99	37.1%	80	30.0%	8	3.0%	55	20.6%	0.138
(m) Clingy behaviour (e.g., following spouse around)	11	4.1%	80	30.0%	81	30.3%	12	4.5%	83	31.1%	0.698
(n) Trouble finding the right word or misnaming things	32	11.8%	150	55.4%	31	11.4%	4	1.5%	54	19.9%	0.123
(o) Feeling paranoid or suspicious	43	15.8%	114	41.9%	46	16.9%	5	1.8%	64	23.5%	0.649

* *p* < 0.05; The symbol “+” means the younger groups were significantly more likely to agree strongly with the item.

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
