# Peer review of "Perceptions of Chinese Towards Dementia in Hong Kong—Diagnosis, Symptoms and Impacts"

_ijerph, 2019, doi:10.3390/ijerph16010128_

Round 1

Reviewer 1 Report

Summary 

As the authors state, Dementia is one of the most significant medical conditions facing the global community. While there is significant research on the overall epidemiology, taxonomy and management strategies,  there is much less on different population perceptions and beliefs. Given the global number of people of Chinese origin it is clearly important to obtain information from them about such matters. 

This paper is thus timely and provides significant additional information about perceptions relating to the impact of dementia on Chinese in Hong Kong. 

Introduction.  The introduction provides a useful summary of the increasing prevalence and impact of dementia from a range of countries and outlines the importance of Hong Kong within a global context as the country with the world's highest life expectancy. 

There is an appropriate outlining of the rationale for the research describing the different community perceptions of receiving knowledge about dementia, and the relative dearth of that information about Chinese populations. There is also an important reminder about the way that different cultural values may impact on practical management implications such as screening initiatives and stigma reduction. 

Materials and Methods. 

The use of a survey questionnaire  is appropriate to answer the research questions and the researchers have used a previously designed instrument. 

Line 78 It would be helpful to have confirmed in the text that the questionnaire had sufficient validation testing to be used in this population.  

Line 85. Further detail about the rationale for the sample selection would be helpful. Were all hospital outpatient clinics sampled, or from specific disciples? Was the sampling frame designed to provide a balance from particular age groups? 

Results. The results are clearly presented using a mixture of text and Tabulation. 

Line 102 What was the decline rate for the questionnaire ? 

Useful information is provided comparing the impact of previous knowledge of someone with dementia in terms of problems encountered within the health system. 

The range of questions and responses provides a useful response set for both clinical management and systems planning. 

Discussion 

The discussion is provided in a logical sequence that places the main findings of the study within existing literature. 

The discussion raises important questions about the underlying values inherent in any particular society (Line 201), and their impact on both care giving and health service provision. 

The discussion appropriately raises the limitation of using a hospital based sample and the potential selection bias in using patients who already have a range of 'pathology'. 

Author Response

Response to Reviewer 1 Comments:

Point 1: Materials and Methods. Line 78. It would be helpful to have confirmed in the text that the questionnaire had sufficient validation testing to be used in this population.  

RESPONSE 1: Thanks for your comment. We have added in the Methods that apart from forward and backward translations, the final questionnaire was pilot-tested for its face- and content-validity with laymen. Most of the items were rated as comprehensible and relevant.

Point 2: Line 85. Further detail about the rationale for the sample selection would be helpful. Were all hospital outpatient clinics sampled, or from specific disciples? Was the sampling frame designed to provide a balance from particular age groups? 

RESPONSE 2: We have added that the outpatients were recruited from an internal medicine department of a regional hospital. Patients at the clinic waiting area of the internal medicine department were invited by research assistants to complete the questionnaire, with inclusion and exclusion criteria applied. The respondents’ age ranged from 18 – 91, with a mean 53.2 (13.24) years. While we did not intend to recruit an even distribution of age groups, the respondents were grouped into three age-groups (<40, 40–64, ≥65 years old) in the analyses.

Point 3: Results. Line 102 What was the decline rate for the questionnaire? 

RESPONSE 3: About 2 out of every 10 eligible subjects rejected to participate, giving a response rate over 75%. We have added the response rate in the results section.

Reviewer 2 Report

It was my plessure to review this manuscript, which is relevant and interesting do deal with dementia. I believe that psychoeducation for caregivers and education to the global society have to be developed after a good understand the general public’s perception and reception of the syndrome.

I included some suggestions to improve the quality of the paper. Please check it.

Author Response

Response to Reviewer 2 Comments:

Point 1: Introduction. For the sentence “Dementia, a chronic disease characterized by…..”, better call it syndrome.

RESPONSE 1: We have amended the wording from “disease” to “syndrome”.

Point 2: It is expected that authors indicate here some sensory, non-pharmacologic interventions, as stated by the best practices statements

RESPONSE 2: We have added in the introduction that there is currently limited evidence for non-pharmacological interventions for people with dementia to effectively slow down decline and delay institutionalisation. However, psychoeducation for caregivers that started earlier in the disease course could be effective in improving caregivers' mood and quality of life.

Point 3: Regarding the Institutional Review Board for ethics approval, is this decision accepted and usual in the country? Is this issue included in the process to ask the permission to the Ethical Committee?

RESPONSE 3: Yes, the Institutional Review Board of The University of Hong Kong /Hospital Authority Hong Kong West Cluster is an official ethics review board in Hong Kong. We have got the approval (UW 18-194).